# Natural Compounds Regulate Macrophage Polarization and Alleviate Inflammation Against ALI/ARDS

**DOI:** 10.3390/biom15020192

**Published:** 2025-01-29

**Authors:** Zhenhuan Yin, Ruizhe Song, Tong Yu, Yunmei Fu, Yan Ding, Hongguang Nie

**Affiliations:** Department of Stem Cells and Regenerative Medicine, College of Basic Medical Science, China Medical University, Shenyang 110122, China; 2023110014@cmu.edu.cn (Z.Y.); 2023120035@cmu.edu.cn (R.S.); 2020110024@stu.cmu.edu.cn (T.Y.); ymfu@cmu.edu.cn (Y.F.); yding@cmu.edu.cn (Y.D.)

**Keywords:** acute lung injury/acute respiratory distress syndrome, natural compound, flavonoid, macrophage polarization, inflammation

## Abstract

Acute lung injury (ALI)/acute respiratory distress syndrome (ARDS) is a pulmonary disease with high mortality associated with inflammation. During the development of ALI/ARDS, macrophages usually polarize toward M1 pro-inflammatory macrophages, promoting the inflammatory response in ALI/ARDS and aggravating lung tissue damage. Natural compounds with anti-inflammatory activity have achieved excellent results in the treatment of ALI/ARDS through different regulatory modes, including macrophage polarization. Of note, flavonoid, brevilin A, and tetrahydropalmatine play an important role in the treatment of ALI/ARDS by modulating the phenotypic polarization of macrophages and their pro-inflammatory cytokine expression in innate immune cells of the lung. Flavonoids are a kind of naturally occurring polyphenol compound, which has antioxidant and anti-inflammatory activities. Studies have found that some flavonoids can alleviate ALI/ARDS through inhibiting the expression of inflammatory cytokines in macrophages. Among them, 5-methoxyflavone, acacetin, grape seed proanthocyanidins, and luteolin can also regulate macrophage polarization. Therefore, the in-depth exploration of the regulatory mechanism of macrophages can lay the foundation for the application of flavonoids in alleviating inflammation-related lung injury. This review focuses on the macrophage polarization effects of different natural compounds and their potential anti-inflammatory mechanisms in the treatment of ALI/ARDS.

## 1. Introduction

Acute lung injury (ALI)/acute respiratory distress syndrome (ARDS) is an acute diffuse inflammatory lung injury caused by shock, trauma, pneumonia, etc., and the fatality rate reaches 30% to 50% [1,2,3,4,5]. ALI/ARDS is usually accompanied by pulmonary edema [6,7] and inflammatory cell infiltration [8,9]. However, at present, the effective treatments for ALI/ARDS are very limited [10,11]. The progress of ALI/ARDS is mainly driven by inflammatory response, evidenced by the activation of immune cells in ALI/ARDS, including macrophages, which secrete inflammatory mediators such as interleukin (IL)-1β, IL-6, IL-18, and tumor necrosis factor (TNF)-α, promoting the injury of alveolar epithelial cells and vascular endothelial cells [12,13,14,15,16,17].

Of note, macrophages are common innate immune cells, among which the alveolar macrophage (AM)-mediated inflammatory response plays a key role in ALI/ARDS [18,19,20,21,22,23]. Under different stimuli, AMs can be polarized into classical activated macrophages (M1) and alternatively activated macrophages (M2) [21,24]. Between them, M1 macrophages produce inflammatory mediators such as TNF-α, IL-1β, and IL-6, which induce the inflammatory response and lung injury in ALI/ARDS [20,25,26]. On the contrary, M2 macrophages secrete anti-inflammatory cytokines such as transforming growth factor-β and IL-10 to inhibit the inflammatory response and participate in tissue repair [25,27]. Bacterial lipopolysaccharide (LPS), as an endotoxin derived from the outer membrane of Gram-negative bacteria, has been widely used to establish inflammatory models in the related studies of ALI/ARDS [4,8,24,28]. The expression of inflammatory mediators in the lungs of ALI induced by LPS increases significantly, while the expression of anti-inflammatory cytokines decreases, accompanied by an up-regulation in the ratio of M1/M2 macrophages [29,30]. In addition, mediators secreted by LPS-stimulated AMs impair the ion transport function of the epithelial sodium channel in distal lung epithelial cell, and have a harmful effect on the ability of the alveolar epithelium to clear edema fluid [31]. Therefore, regulating the polarization of macrophages and accompanied inflammation could be an effective strategy for the treatment of ALI/ARDS [20,26].

In recent years, natural compounds have shown promising prospects in therapeutic studies for a variety of inflammation-related diseases, among which flavonoids play an important role in treating LPS-induced ALI [32,33,34,35,36]. This review focuses on natural compounds that modulate the phenotypic polarization of macrophages and pro-inflammatory cytokine expression, which will prove to be a great help for the mechanism exploration of ALI/ARDS.

## 2. Therapeutical Effect of Natural Compounds in ALI/ARDS

A variety of natural compounds can regulate the polarization and inflammatory response induced by macrophages by acting on macrophages, which have proved to be a great help for the exploration of the mechanism in the treatment of ALI/ARDS [32,37,38]. In this part, natural compounds are classified and summarized according to their types.

### 2.1. Diterpenoid

Andrographolide, a diterpenoid lactone derived from *Andrographis paniculata*, has been found to have an anti-inflammatory effect [39,40]. Abietic acid is another diterpenoid among the terpenoids isolated from *Pimenta racemosa* var. grissea that has anti-inflammatory and antioxidant effects [41,42].

As a new natural andrographolide product, 3-dehydroandrographolide and abietic acid can effectively reduce the inflammatory response induced by macrophages in ALI, in which abietic acid can inhibit the polarization of macrophages to M1 and reduce the expression of pro-inflammatory cytokines in ALI/ARDS induced by LPS [43,44].

### 2.2. Flavonoid

Flavonoids are a kind of naturally occurring polyphenolic compound, widely found in fruits, vegetables, and herbs, which have antioxidant and anti-inflammatory activities [45]. Of note, flavonoids play a significant role in the treatment of ALI/ARDS through anti-inflammation, macrophage regulation, etc. [32,37,46,47,48].

5-Methoxyflavone is a flavonoid with DNA polymerase-β inhibitory properties that can inhibit the infiltration of inflammatory cells, alveolar edema, and cell apoptosis by decreasing the expression of inflammatory cytokines such as IL-6 and TNF-α in lung tissues, thus reducing the lung index and lung injury score of mice and alleviating LPS-induced ALI [45,49].

Moreover, naringin, baicalein, hydnocarpin D, myricetin, acacetin, grape seed proanthocyanidins, and luteolin are also natural flavonoids derived from plants with anti-inflammatory and antioxidant functions [32,38,46,50,51,52,53,54,55,56]. Research has found that most of these flavonoids can inhibit inflammatory cell infiltration and decrease the expression of inflammatory factors TNF-α, IL-6, and IL-1β, thereby reducing alveolar permeability and relieving alveolar edema in ALI [38,49,50]. In addition, naringin can effectively inhibit the expression of pro-inflammatory cytokines of macrophages [57]. Notably, luteolin significantly reduces the expression level of ROS in lung tissue, increases the expression of IL-10, and promotes the polarization of macrophages to M2, all of which can improve the survival rate of LPS-induced ALI/ARDS in mice [32,58].

### 2.3. Sesquiterpene Lactone

Brevilin A is an anti-inflammatory sesquiterpene lactone extracted from *Centipeda minima* [59,60]. Costunolide, parthenolide, and isoalantolactone are also sesquiterpene lactones extracted from plants, and all of them have anti-inflammatory effects [61,62,63,64,65]. It has been found that these compounds can effectively reduce the secretion of pro-inflammatory cytokines of macrophages [59,63,65,66]. Among them, brevilin A can promote the polarization of macrophages to M2 [59].

Dehydrocostus lactone is another natural sesquiterpene lactone extracted from the traditional Chinese herbs *Saussurea lappa* and *Inula helenium* L. with anti-inflammatory effects [67,68]. Dehydrocostus lactone can inhibit inflammation in ALI/ARDS induced by methicillin-resistant *S. aureus* by promoting the polarization of M2 macrophages and inhibiting the M1 polarization of macrophages [68].

### 2.4. Alkaloid

Capsaicin, a natural alkaloid found in chili pepper, has the same anti-inflammatory effects as alkaloids from plants such as nuciferine, isorhynchophylline, tabersonine, and isocorydin [69,70,71,72]. Nanoparticles containing capsaicin and iron (Fe-CAP NPs), constructed by the combination of capsaicin and iron, and other drugs mentioned above could effectively inhibit the secretion of pro-inflammatory cytokines of macrophages, and had a certain protective effect on lung tissue in ALI/ARDS [69,70,71,72,73].

In addition, tetrahydropalmatine is also a natural alkaloid derived from the traditional Chinese medicine *Corydalis yanhusuo*, which has antioxidant and anti-inflammatory effects [33,74]. Tetrahydropalmatine can promote the polarization of M1 macrophages to M2 in ischemia-reperfusion-induced ALI and inhibit the inflammatory response in ALI [33].

### 2.5. Isoflavone

Formononetin is an isoflavone derived from soy, red clover, and other plants, and has antioxidant and anti-inflammatory properties [75,76]. Sophoricoside is also an anti-inflammatory isoflavone glycoside isolated from the seed of *Sophora japonica* L. [77,78]. Formononetin has a protective effect on lung injury by reversing the inhibition of the polarization of M2 macrophages in hyperoxia-induced ALI, and sophoricoside can effectively inhibit the secretion of pro-inflammatory cytokines in macrophages induced by LPS, thus alleviating the inflammatory response in ALI/ARDS [75,77].

### 2.6. Anthraquinone

Rhein is an anthraquinone derived from many traditional Chinese medicines, and has antioxidant and anti-inflammatory properties [36,79]. It can promote the M2 polarization of macrophages and effectively inhibit inflammation in ALI/ARDS induced by LPS [36].

Moreover, *Ephedra* is a somewhat toxic medicinal plant with antioxidant and anti-inflammatory properties, and the dried stems and leaves of *Ephedra* are prescribed in China and Japan for the treatment of coughs and asthma [80,81]. As one of the main components of *Ephedra*, pseudoephedrine can act directly on α-adrenergic receptors and has significant anti-inflammatory effects. It has been found, in a study, that emodin combined with pseudoephedrine can inhibit the polarization of M1 macrophages in ALI/ARDS induced by LPS, promote the polarization of M2 macrophages, and reduce the secretion of pro-inflammatory cytokines [80].

Moreover, emodin is also a natural anthraquinone derivative with anti-inflammatory functions isolated from traditional Chinese medicines, including *Rheum palmatum* L., *Rheum officinale*, and other plants [82,83]. Emodin can significantly inhibit the inflammatory response induced by macrophages and decrease the expression of pro-inflammatory cytokines in lung tissue, sequentially improving lung tissue injury in ALI [83].

### 2.7. Triterpenoid Saponin

Hederagenin is a triterpenoid saponin derived from *Hedera helix* and other plants that has an anti-inflammatory function [84,85,86]. A previous report showed that hederagenin could inhibit LPS-induced M1 polarization and the pro-inflammatory cytokine expression of macrophages in vivo and in vitro, and improve LPS-induced lung tissue injury and inflammatory cell infiltration in ALI [87].

Ginseng has been one of the important herbs in traditional Asian medicine for thousands of years [88,89]. Ginsenoside Rb1 is the main component of ginsenosides, which has antioxidative and anti-inflammatory functions [34,90]. Moreover, hederasaponin C is a triterpenoid saponin derived from the traditional Chinese medicine *Pulsatilla chinensis* (Bunge) Regel [91]. Studies have shown that ginsenoside Rb1 and hederasaponin C can inhibit the expression of pro-inflammatory cytokines of macrophages in ALI/ARDS [34,91].

### 2.8. Biphenolic

Curcumin is a biphenolic compound derived from the rhizome of turmeric (*Curcuma longa*), which has antioxidant and anti-inflammatory properties [92,93]. It can suppress inflammation by inhibiting the LPS-induced production of IL-8, monocyte chemoattractant protein-1, macrophage inflammatory protein-1α, and TNF-α by AM [94].

However, due to curcumin’s instability and poor solubility in water, leading to its low oral bioavailability, it is difficult to achieve the minimum effective therapeutic concentration in the clinic, which is thought to be related to the active methylene groups of the β-diketone moiety in the structure of curcumin [95]. Piperlongumine is a compound with anticancer and anti-inflammatory activities in plant long pepper (*Piper longum* L.). A study was conducted on piperidin-2-one and 3,4,5-trimethoxyphenyl moieties from piperlongumine into the curcumin skeleton to construct a series of novel di-carbonyl analogs of curcumin (DACs), and the results showed that DACs had higher chemical stability than curcumin and can inhibit the expression of LPS-induced pro-inflammatory cytokines in macrophages, which may be an effective strategy for the treatment of ALI/ARDS [95].

### 2.9. Chalcone

Cardamonin is a naturally occurring chalcone compound isolated from plants with anti-inflammatory and antioxidant activities [10,96]. In addition, flavokawain B, another natural chalcone isolated from the root extract of *Piper methysticum*, also has significant anti-inflammatory activity [35]. Cardamonin and flavokawain B can both inhibit the secretion of pro-inflammatory cytokines in macrophages induced by LPS, and alleviate pulmonary inflammation in ALI [10,35].

## 3. Regulating Macrophages and Inflammatory Response via Natural Compounds in ALI/ARDS

Macrophages in ALI/ARDS are regulated by multiple pathways during their participation in inflammatory responses. For example, nuclear factor (NF)-κB and mitogen-activated protein kinase (MAPK) signaling pathways play a key role in macrophage polarization and the expression of inflammatory cytokines [35,59,97,98,99]. At the same time, some natural compounds exert their protective effects in inflammation-related ALI/ARDS by regulating macrophage polarization. Among them, the regulatory effects and mechanisms of some natural compounds are summarized in Table 1; most of their chemical structures are also seen in the Appendix A.

### 3.1. NF-κB and MAPK Signaling Pathways

NF-κB and MAPK signaling pathways promote the transcription of downstream inflammatory genes and the expression of pro-inflammatory cytokines, such as TNF-α, IL-6, and NOD-like receptor protein 3, and the M1 polarization of macrophages [30,35,59,87,97,98,99]. The MAPK signaling pathway consists of c-Jun NH2-terminal protein kinases (JNK), p38, and extracellular signal-related protein kinases (ERK), which are activated under LPS stimulation [29,65,119]. When the inhibitor of κB kinase α/β (IKKα/β) promotes IκB phosphorylation and degradation, NF-κB p65 is transported from the cytoplasm to the nucleus to induce the expression of inflammatory genes [43,59]. By inhibiting the phosphorylation of MAPK and NF-κB signaling pathways in macrophages, it is possible to effectively suppress the expression of inflammatory factors in LPS-induced macrophages and alleviate ALI [34,44,54,69,120].

A previous study showed that brevilin A could bind with the inhibitor of κB kinase α/β to inhibit its phosphorylation, reduce the activation of the NF-κB pathway in macrophages induced by LPS/IFN-γ in a dose-dependent manner, decrease the expression of pro-inflammatory cytokines such as IL-1β, IL-6, and TNF-α in macrophages, and effectively alleviate inflammatory reaction and lung tissue injury in ALI/ARDS [59].

TAK1 is a highly conserved MAPK kinase, which can promote the expression of pro-inflammatory cytokines when it binds to phosphatidylinositol-4,5-bisphosphate (PIP2), while LPS stimulation can increase the interaction between TAK1 and endogenous PIP2 in macrophages [91]. Hederasaponin C can inhibit the activation of NF-κB and NLRP3 inflammasome by inhibiting the binding of TAK1 and PIP2 in macrophages induced by LPS, thus inhibiting ALI induced by LPS [91].

Hederagenin, 3-dehydroandrographolide, Fe-CAP NPs, and abietic acid can inhibit the expression of inflammatory mediators, such as IL-6 and TNF-α in macrophages, by inhibiting the NF-κB signal pathway. Among them, 3-dehydroandrographolide regulates the NF-κB/Akt signal pathway mainly by activating the cholinergic anti-inflammatory pathway; in addition, hederagenin and abietic acid can inhibit the polarization of macrophages to M1, while Fe-CAP NPs can increase the expression of anti-inflammatory cytokine TGF-β, but it is not clear whether it can regulate the phenotype of macrophages in the process of anti-inflammation [43,44,69]. Costunolide can reduce the expression of pro-inflammatory cytokines such as IL-6, TNF-α, and IL-1β in macrophages induced by heat-killed S. aureus by inhibiting the MAPK signal pathway, which has a good therapeutic effect on septic lung injury induced by heat-killed S. aureus [61].

By inhibiting the MAPK and NF-κB signaling pathways, ginsenoside Rb1 can reduce the inflammatory response induced by macrophages in Staphylococcus aureus-induced ALI, while DACs and isocorydine can alleviate inhibit the expression of pro-inflammatory cytokines of macrophages in ALI/ARDS induced by LPS [34,71,95].

Hydnocarpin D, naringin, and myricetin can inhibit the activation of the NF-κB signaling pathway by inhibiting the phosphorylation and nuclear translocation of NF-κB p65, and downgrade the activation of the MAPK signaling pathway by suppressing the phosphorylation of ERK, JNK, and p38, thereby blocking the expression of pro-inflammatory cytokines in macrophages in ALI (Figure 1A–C) [54,57,120].

In addition, toll-like receptor 4 (TLR4) is an important member of the toll-like receptor family, and the TLR4-NF-κB/MAPK signaling pathway can induce the M1 polarization of macrophages [99,114]. LPS can activate TLR4 receptors and bind to TLR4 co-receptor myeloid differentiation protein 2, thereby activating the downstream adapter myeloid differentiation primary response gene 88 for signal transduction [10,35,73,95,99,100]. Meanwhile, TNF receptor-associated factor 6 (TRAF6) is a downstream signaling molecule of TLR4-myeloid differentiation primary response gene 88, the ubiquitination of which can promote the activation of the NF-κB and MAPK signaling pathways, while the loss of TRAF6 can induce the M2 polarization of macrophages and alleviate the inflammatory response in ALI [37,61,71,72].

Flavokawain B, nuciferine, isorhynchophylline, and tetrahydropalmatine can down-regulate the inflammatory response caused by macrophages in ALI and alleviate lung injury in ALI by inhibiting the TLR4/NF-κB signal pathway. Among them, flavokawain B can inhibit the activation of the NF-κB and MAPK signal pathway by inhibiting the formation of the LPS/MD2/TLR4 complex, and tetrahydropalmatine can induce M1 macrophages to polarize to M2 by inhibiting the TLR4/NF-κB/NLRP3 signal pathway, thus alleviating the inflammatory response in ischemia-reperfusion-induced ALI [33,35,70,73].

Isoalantolactone and tabersonine can inhibit TRAF6 and thus inhibit the activation of the NF-κB signal pathway, and tabersonine can also inhibit the activation of the MAPK signal pathway and reduce the expression of macrophage pro-inflammatory cytokines in ALI induced by LPS, showing significant anti-inflammatory activity in ALI [65,72]. Acacetin can promote the M2 polarization of macrophages by inhibiting the TRAF6/NF-κB/COX-2 signal pathway and alleviate lung tissue damage in ALI induced by sepsis (Figure 1D) [37].

### 3.2. Nuclear Factor Erythroid 2-Related Factor 2/Heme Oxygenase-1

The nuclear factor erythroid 2-related factor 2 (Nrf2) is a key transcription factor that is normally inactivated by the interaction with its negative regulatory factor kelch-like ECH-associated protein 1 (Keap1). Stimuli such as oxidative stress release Nrf2 from keap1 undergoes nuclear translocation and binds to the antioxidant response element, thereby inducing the expression of the downstream antioxidant protein heme oxygenase-1 (HO-1) [66,121,122], the latter of which is related to the down-regulation of NF-κB activation. Therefore, activating the Nrf2/HO-1 signaling pathway can effectively alleviate the inflammatory response in ALI [29,45,66,77,123].

Isolinderalactone and formononetin can alleviate effectively ALI/ARDS by activating the Nrf2 signal pathway in macrophages, in which isolinderalactone can inhibit the NF-κB signal pathway in macrophages while activating the Nrf2 signal pathway, and formononetin can reverse the decrease in the polarization of M2 macrophages in hyperoxia-induced ALI [66,75].

Studies show that 5-methoxyflavone and hydnocarpin D can significantly regulate the inflammatory response in ALI by reducing the expression of Keap1 in macrophages and promoting the expression of antioxidant factors Nrf2 and HO-1, in which 5-methoxyflavone can inhibit M1 polarization of macrophages in ALI and M1 repolarization of M2 macrophages (Figure 2A,B) [45,54]. In addition, luteolin promotes the ERK1/2 signaling pathway, Ca^2+^ efflux, and HO-1 expression in macrophages and inhibits the inflammatory response to alleviate ALI, which also promotes macrophage polarization to the M2 type in ALI by promoting regulatory T cell differentiation for IL-10 expression [32,121].

### 3.3. Phosphatidylinositol-3 Kinase/Protein Kinase B

In recent years, the phosphatidylinositol-3 kinase (PI3K)/protein kinase B (Akt) signaling pathway has been a hot research topic in the treatment of ALI, the former of which is a member of the phosphatidylinositol-3 kinase family [124,125]. The triggering receptor expressed on myeloid cells 2 is a major member of the super immunoglobulin family triggering receptors expressed on myeloid, whose activation can promote the phosphorylation of the PI3K/Akt signal pathway downstream, regulate the polarization of macrophages from M1 to M2, and reduce LPS-induced ALI by binding to grape seed proanthocyanidin (Figure 1E) [38].

Baicalein alleviates ALI by inhibiting the expression of dynamin-related protein 1 (Drp1) with the excessive fission of mitochondrial fission, and effectively mitigates mitochondrial mass in LPS-activated macrophages [46,126]. Other studies have shown that enhancing mitochondrial oxidative phosphorylation in macrophages and inhibiting glycolysis can promote macrophage M2 polarization and reduce the expression of inflammatory cytokines [127,128].

### 3.4. AMP-Activated Protein Kinase

As a sensor of intracellular energy metabolism, AMP-activated protein kinase (AMPK) is used to regulate acute or chronic inflammation in various cells and animal models, which can inhibit the degradation of IκBα and the expression of pro-inflammatory cytokines such as TNF-α and IL-6 in macrophages induced by LPS [94,117,122]. The activation of AMPK and peroxisome proliferator-activated receptor γ can significantly reduce the expression of inflammatory mediators and tissue damage in sepsis and significantly promote the polarization of macrophages to M2 phenotype [26].

Dehydrocostus lactone can inhibit the inflammatory response induced by methicillin-resistant *S. aureus* by inhibiting MAPK and NF-κB signals and activating the AMPK/Nrf2 pathway in vitro and in vivo, and promote macrophages to polarize from M1 to M2 to inhibit ALI [68]. Moreover, curcumin can also inhibit the NF-κB signal pathway by activating the AMPK signal pathway, reduce the production of LPS-induced TNF-α and IL-6 pro-inflammatory cytokines in macrophages, and alleviate LPS-induced lung injury and inflammation [94]. In addition, sophoricoside can inhibit the production of iNOS, NO, TNF-α, IL-1β, and IL-6 in macrophages induced by LPS by activating the AMPK/Nrf2 pathway and improve the inflammatory response and pathological changes in lung tissue induced by LPS [77].

### 3.5. Mammalian Target of Rapamycin

The activation of the mammalian target of the rapamycin signaling pathway can promote the translocation of hypoxia-inducible factor-1α from cytoplasm to nucleus, increase the expression of vascular endothelial growth factor, and increase the infiltration of pro-inflammatory cytokines, which plays an important role in the disease progression of ALI [83]. Emodin can down-regulate the expression of hypoxia-inducible factor-1α and vascular endothelial growth factor genes by inhibiting the mammalian target of the rapamycin signal pathway, thus inhibit the expression of pro-inflammatory cytokines such as TNF-α, IL-1β, and IL-6 in macrophages and lung tissue induced by LPS, and can improve the inflammatory response in ALI/ARDS [83].

### 3.6. Vasoactive Intestinal Peptide/Cyclic Adenosine Monophosphate/Phosphorylate Protein Kinase

Vasoactive intestinal peptide (VIP) is one of the important immunoreactive neuropeptides in the lungs that can inhibit granulocyte recruitment and promote the expression of anti-inflammatory factors IL-4 and IL-10. It has been found that VIP can activate cyclic adenosine monophosphate (cAMP) to phosphorylate protein kinase (PKA) [80]. The results of studies show that the combination of emodin and pseudoephedrine to activate the VIP/cAMP/PKA signal pathway can inhibit NF-κB and the MAPK signal pathway in ALI induced by LPS, inhibit the polarization of M1 macrophages, promote the polarization of M2 macrophages, and reduce the expression of pro-inflammatory cytokines in ALI/ARDS [80].

## 4. Clinical Application of Natural Compounds in ALI/ARDS-Related Diseases

ARDS associated with coronavirus disease 2019 (COVID-19) has been a challenge in intensive care medicine for the past three years; a respiratory system failure mainly caused by the accumulation of lung inflammatory cytokines [129,130]. Severe immune responses are associated with the mortality rate of COVID-19 patients [131,132]. *Yindan Jiedu* granules is a compounded Chinese herbal medicine composed of Maxing Shigan and Qingwen Baidu decoction that can shorten the average duration of fever and pulmonary exudative lesions in mild/moderate COVID-19 patients. Both network pharmacology and experimental analysis identified that luteolin, quercetin, and kaempferol, as the main constituents of Yindan Jiedu granules, may improve the inflammatory response of COVID-19 patients by regulating inflammatory pathways [133]. In addition, Shufeng Jiedu capsule is a patented herbal medicine composed of eight medicinal plants for the treatment of different viral respiratory infectious diseases, the early application of which within the first 8 days from the onset of COVID-19 has shown a remarkable effect [134]. As a promising prophylactic, and therapeutic candidate for COVID-19 patients, curcumin is isolated from traditional medicines, which may alleviate the disease by regulating the inflammatory response and preventing the progression of tissue damage. Moreover, nano-curcumin prepared by nanotechnology may cause an improvement in clinical manifestation including fever, cough, and dyspnea and overall recovery in COVID-19 patients by modulating the increased rate of inflammatory cytokines [135,136]. As a flavonoid compound, quercetin, when combined with antiviral drugs, can relieve clinical symptoms and shorten hospital stay compared with using antiviral drugs alone, which is safe and effective in lowering the serum levels of critical markers involved in COVID-19 severity [137]. Moreover, epigallocatechin-3-gallate, another representative flavonoid component, the aerosol inhalation of which is well tolerated, and has been reported to be effective in controlling the progress and promoting the improvement of COVID-19-related interstitial pneumonia in cancer patients through a phase I–II clinical trial [138]. In addition, a double-blinded, randomized, and placebo-controlled trial showed that, although colchicine, a compound in an alkaloid, was not found to have an early beneficial effect on reducing mortality, a delayed beneficial effect was observed in alleviating the exacerbation of COVID-19 patients [139].

Ginsenosides, the main active ingredients of ginseng saponins, are steroid compounds with high anti-tumor, anti-shock, and anti-inflammatory activity that can improve and strengthen immunity. The synergistic use of ginsenosides and ulinastatin can strengthen the sole effect of ulinastatin in improving patients’ pulmonary vascular permeability, pulmonary circulation, blood gases, and hemodynamics, as well as APACHE II and ALI scores in patients with ALI/ARDS caused by sepsis, a main cause of death clinically [140].

## 5. Conclusions

Natural compounds have been widely studied and applied in the treatment of diseases including ALI/ARDS. The mechanisms of regulating macrophage polarization and inhibiting inflammatory response have come to the fore. In recent years, natural compounds play a significant role in the treatment of ALI/ARDS through their regulatory effects on macrophages. Recent studies have shown that natural compounds inhibit the inflammatory response induced by macrophages by regulating the polarization of macrophages through a variety of mechanisms, including classic pathways such as NF-κB and MAPK. Of note, luteolin can indirectly induce the polarization of macrophages to the M2 type by promoting the expression of IL-10 in regulatory T cells in lung tissues, which also indicates that, besides directly regulating macrophages, reconciling the interaction of immune cells is of great significance in alleviating the inflammatory response in ALI. The in-depth study of the mechanism of action of flavonoids, and the exploration of the combined use of different natural compounds, can further advance the study of the mechanisms of macrophage regulation in ALI/ARDS therapy.

## Figures and Tables

**Figure 1 biomolecules-15-00192-f001:**
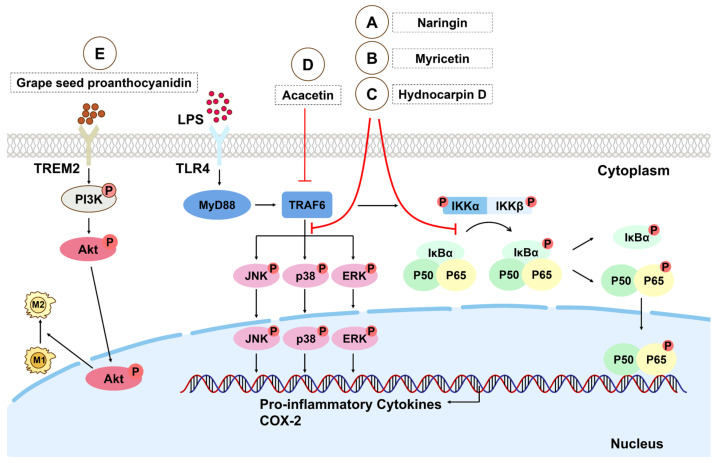
Regulatory mechanism of flavonoids on macrophages in LPS stimulation. (**A–C**) Naringin, myricetin, and hydnocarpin D attenuate the activation of the NF-κB and MAPK signal pathways by inhibiting the degradation of IκBα, nuclear translocation of NF-κB p65, and phosphorylation of ERK, JNK, and p38 in macrophages, thus blocking the expression of inflammatory factors in macrophages. (**D**) Acacetin can inhibit the TRAF6/NF-κB/COX-2 signal pathway. (**E**) The activation of TREM2 by binding to grape seed proanthocyanidin can promote the phosphorylation of the PI3K/Akt signal pathway downstream and regulate the polarization of macrophages from M1 to M2. Akt: protein kinase B; COX-2: cyclooxygenase-2; ERK: extracellular signal-regulated kinase; Ikkα/β: nuclear factor κB kinase α/β; IκB: inhibitor of NF-κB; JNK: c-Jun N-terminal kinases; MAPK: mitogen-activated protein kinase; MyD88: myeloid differentiation factor-88; NF-κB: nuclear factor-κB; PI3K: phosphoinositide 3-kinase; TRAF: TNF receptor-associated factor; TLR4: toll-like receptor 4; TRAF6: TNF receptor-associated factor 6; TREM2: triggering receptor expressed on myeloid cells 2.

**Figure 2 biomolecules-15-00192-f002:**
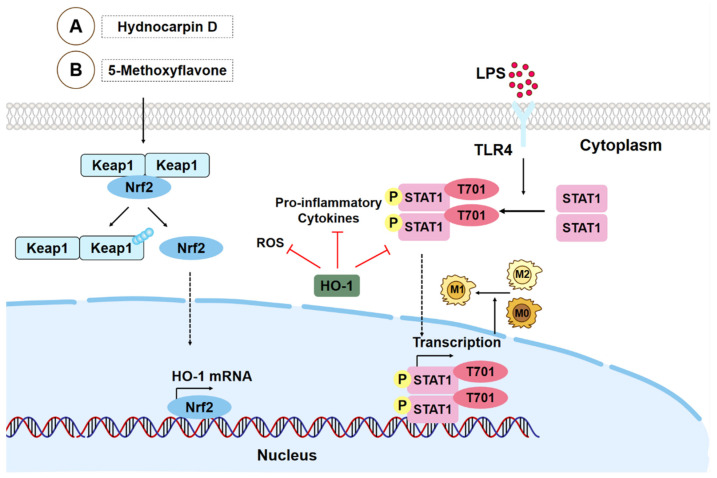
Effects of hydnocarpin D and 5-methoxyflavone on macrophages in LPS stimulation. (**A**,**B**) Hydnocarpin D can increase the expression of antioxidant factors Nrf2 and HO-1 and reduce the expression of its negative regulatory factor Keap1, thus inhibiting the inflammatory response and oxidative stress caused by macrophages. 5-Methoxyflavone inhibits the activation of STAT1 signal by activating the Nrf2/HO-1 signal pathway in macrophages and, finally, reverses the polarization of M1 macrophages and the repolarization of M2 macrophages to M1. HO-1: heme oxygenase-1; Keap1: kelch-like ECH-associated protein 1; Nrf2: nuclear factor erythroid 2-related factor 2; ROS: reactive oxygen species; STAT1: signal transducer and activator of transcription-1.

**Table 1 biomolecules-15-00192-t001:** Regulatory mechanisms of natural compounds on macrophages in ALI/ARDS.

Compound Name	Compound Type	Effect on Macrophages in ALI/ARDS	Molecular Mechanism	Refs.
2-Hydroxymethyl anthraquinone	Anthraquinone	Inhibition of NO, TNF-α, IL-6, and IL-1β, expression induced by LPS	Inhibition of TLR4/NF-κB signal pathway	[100]
Loganin	Iridoid glucoside	Inhibition of M1 polarization induced by LPS and inhibition of TNF-α and IL-6 expression	Inhibition of ERK/NF-κB signal pathway	[98,101,102,103]
Smiglaside A	Phenylpropanoid glycoside	Inhibition of LPS-induced M1 polarization and promotion of M2 polarization	Activation of AMPK/PPARγ signal pathway	[26]
Salidroside	Phenylpropanoid glycoside	Inhibition of TNF-α and IL-6 expression induced by LPS	Inhibition of NF-κB and NLRP3 signaling pathways	[104,105,106]
Ethyl ferulate	Phenylpropanoid	Inhibition of LPS-induced pro-inflammatory cytokine secretion	Activation of Nrf2/HO-1 signal pathway and inhibition of NF-κB signal pathway	[97,107,108]
Resveratrol	Polyphenolic	Promotion of M2 polarization and inhibition of LPS-induced expression of IL-6 and IL-1β	Activation of STAT3/SOCS3 signal pathway	[109,110,111]
Epigallocatechin-3-gallate	Polyphenolic	Promotion of M2 polarization and inhibition of LPS-induced expression of iNOS, TNF-α, IL-1β, and IL-6	Activation of KLF4 signal pathway	[112,113]
Polygonatum sibiricum polysaccharides	Polysaccharides	Inhibition of M1 polarization and promotion of M2 polarization, inhibition of LPS-induced expression of TNF-α, IL-1β, and IL-6	Inhibition of TLR4/MAPK/NF-κB signal pathway	[114,115]
Allyl methyl trisulfide	Sulfide	Inhibition of LPS-induced M1 polarization and reduction in the expression of inflammatory mediators	Inhibition of NF-κB and MAPK signaling pathways	[99]
Cryptotanshinone	Diterpenoid quinone	Promotion of polarization from M1 to M2 and inhibition of inflammatory response induced by LPS	Activation of AMPK signal pathway	[116,117]
Rhein	Anthraquinone	Promotion of M2 polarization and reduction in LPS-induced pro-inflammatory cytokine expression	Activation of NFATc1/Trem2 signal pathway	[36,79,118]

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
