# Peer review of "Natural Compounds Regulate Macrophage Polarization and Alleviate Inflammation Against ALI/ARDS"

_biomolecules, 2025, doi:10.3390/biom15020192_

Round 1
Reviewer 1 Report
Comments and Suggestions for Authors
Natural compounds can suppress the inflammatory response triggered by macrophages by influencing their polarization, primarily by modulating classic signaling pathways like NF-κB and MAPK, thereby shifting macrophages towards a less inflammatory phenotype; essentially, these compounds can control the activation state of macrophages to reduce inflammation.
In this review, the authors focused on the macrophage polarization effects of different natural compounds and their potential anti-inflammatory mechanisms in the treatment of acute lung injury(ALI)/acute respiratory distress syndrome(ARDS). Besides a list of known therapeutical effect of natural compounds in ALI/ARDS, the authors also addressed several natural compounds, including flavonoids like naringin, myricetin, hydnocarpin D and acacetin shown to protect against inflammation-related ALI/ARDS by regulating macrophage polarization, primarily by promoting the shift from pro-inflammatory M1 macrophages to anti-inflammatory M2 macrophages, thus mitigating excessive inflammatory responses in the lung tissue; these effects often involve modulation of signaling pathways like NF-κB, TRAF6/COX-2, and PI3K/Akt depending on the specific compound.
The review successfully accomplished the goal of elucidating the mechanisms known of regulating macrophage polarization and inhibiting inflammatory response.
Reviewer 2 Report
Comments and Suggestions for Authors
I was very pleased to read the review by the respected Zhenhuan Yin et al. The review is devoted to the phenomenon of regulation of macrophage polarization by natural compounds in the context of acute lung injury. The strengths of the review are a large volume of the material, a large range of described compounds and an in-depth discussion of the mechanisms, including signaling pathways. Another strength of this review is the possibility of using information not only by scientists, but also by practicing physicians.
The relevance of the topic is due, firstly, to the discussed pathology. COVID-19, as is known, manifested itself in some cases as acute respiratory distress syndrome, leading to death. Despite numerous studies, the pathogenesis of this disease and methods of treatment have not been fully clarified. On the other hand, natural components have very high biological activity, and their use in therapy can significantly improve the effectiveness of treatment.
The review is clearly structured, the main part consists of two subsections. The first subsection describes the main classes and examples of compounds that affect macrophage polarization in acute lung injury. The second subsection focuses on the description of the mechanisms of action. The review fully meets the stated objective and is quite comprehensive.
No similar review has been published in the last ten years or earlier. There have been papers reviewing the role of macrophages in ARDS, but there have been no reviews covering the mechanisms of action of natural compounds in ARDS.
The review includes 141 references, the vast majority of which are from the last five years. Almost all the most important papers are mentioned. Thus, the review contains the most up-to-date information.
The conclusions made by the authors are correct and supported by references.
The review contains two original figures illustrating the mechanisms of polarization of macrophages through the main signaling pathways, as well as a table summarizing the effects of the key natural compounds.
I highly appreciate the work of the authors, but I missed several points that I hope the authors will take into account.
1. As a practicing physician, I would be very interested in information regarding examples of clinical use of the compounds mentioned by the authors for the treatment of patients with ARDS. Subsection 2 contains a lot of information, but it is difficult to extract studies conducted on patients. I would ask the authors to make a separate small subsection, in which clinical studies of the use of substances on patients were summarized.
2. I would ask the authors to present the chemical structures for the most important compounds in the Supplementary file. As I understand it, these are the compounds from Table 1.
Reviewer 3 Report
Comments and Suggestions for Authors
The reviewer has virtually no complaints about the content and design of the material submitted to the editors. The only recommendation: in the text and list of references, the Latin names of the plants from which the active components are extracted should be highlighted in italics.
